# Optimizing Algorithm for Existing Fiber-Optic Displacement Sensor Performance

**DOI:** 10.3390/s24020448

**Published:** 2024-01-11

**Authors:** Zeina Elrawashdeh, Christine Prelle, Frédéric Lamarque, Philippe Revel, Stéphane Galland

**Affiliations:** 1Institut Catholique d’Arts et Métiers (ICAM), Site of Grand Paris Sud, 77127 Lieusaint, France; 2ICB UMR 6303, CNRS, Université Bourgogne Franche-Comté, UTBM, 90010 Belfort, France; 3Royallieu Research Center, Roberval Laboratory (Mechanics, Energy and Electricity), University of Technology of Compiègne, Rue Roger Couttolenc, 60200 Compiègne, France; christine.prelle@utc.fr (C.P.); frederic.lamarque@utc.fr (F.L.); philippe.revel@utc.fr (P.R.); 4Université de Technologie de Belfort Montbéliard UTBM, Laboratoire Connaissance et Intelligence Artificielle CIAD UR 7533, 90010 Belfort, France; stephane.galland@utbm.fr

**Keywords:** fiber-optic sensor, displacement, global optimum, measurement range, resolution, sensitivity

## Abstract

This paper describes the optimal design of a miniature fiber-optic linear displacement sensor. It is characterized by its ability to measure displacements along a millimetric range with sub-micrometric resolution. The sensor consists of a triangular reflective grating and two fiber-optic probes. The measurement principle of the sensor is presented. The design of the sensor’s triangular grating has been geometrically optimized by considering the step angle of the grating to enhance the sensor’s resolution. The optimization method revealed a global optimum at which the highest resolution is obtained.

## 1. Introduction

Highly precise, low-power micro-electro-mechanical system (MEMS)-based devices have been one of the main subjects of research in recent years. The development of micro-sensors with high sensitivity, a large dynamic range, and low power dissipation dominates the research field for various commercial applications, including transportation, biomedicine, space, avionics, and environmental monitoring [1].

High-resolution optical displacement sensors based on Fabry–Perot interferometers have been widely used in MEMS systems due to their high displacement accuracy and immunity to electromagnetic noise [2]. The study conducted by Chung-Ping Chang et al. [3] modified the design of the conventional Fabry–Perot interferometer, enabling a 100 mm measurement range to be achieved with an optical resolution enhanced to a quarter of the wavelength.

In the field of precision nanometrology, Kuang-Chao Fan et al. [4] developed a measurement system that consists of a mini linear diffraction grating interferometer (LDGI) with dimensions of 50 × 30 × 30 mm^3^. The LDGI, together with a focus probe, is integrated into the spindle system of a micro-/nano-coordinate measurement machine (CMM). The sensor delivers an accuracy of 30 nm over the 10 mm displacement range of the spindle. Another example is the wide-range, three-axis grating encoder developed by Jie Lin et al. [5]. This sensor can measure the translational motions of the *x*-, *y*-, and *z*-axes of a stage simultaneously. The grating encoder is composed of a reflective-type planar scale grating with a period of 8 µm and an optical reading head. To make the grating encoder more compact, a double-grating beam-splitting unit and two diffractive optical elements are introduced. The experimental results revealed a resolution of 4 nm for the axial displacement of the *z*-axis.

Akihide Kimura et al. [6] described a three-axis surface encoder consisting of a planar grating and an optical sensor head. It was designed and manufactured for sub-nanometric displacement measurement along the *x*-, *y*-, and *z*-axes. The optical sensor head had dimensions of 50 mm (X) × 70 mm (Y) × 40 mm (Z), and the sensor resolution was better than 1 nm in all three axes.

Several studies based on other optical technologies for linear displacement measurement have also been conducted. A high-performance optical sensor was constructed by A. Missoffe et al. [7]. The compact measurement system consists of a laser diode module along with a photodiode array. This system is characterized by its insensitivity to major mechanical defects. The experimental results showed that the sensor can achieve nanometric resolution over a centimeter travel range.

The actual study considers different aspects. The developed sensor should satisfy several requirements, such as millimetric range, sub-micrometric resolution, and a miniature size.

Fiber-optic technology is a very good candidate because it has several advantages, such as high compactness, low-cost fabrication processes, and compatibility with other optical components, making it an attractive instrument for sensing applications. Intensity-based fiber optic sensors are the earliest and most widely used technology to date due to their low cost, easy installation, and high sensitivity [8]. They could represent a credible alternative to optical micro-encoders when high resolution over long-range measurements are required [9].

The study presented by Tian-Liang et al. [10] illustrates the design of a novel fiber Bragg grating (FBG) displacement sensor. The measurement principle is based on the use of the transverse property of a suspended optical fiber with a pre-tension force. The theoretical model has been derived and validated, and the design has an excellent sensitivity of 2086.27 pm/mm and a high resolution of 0.48 µm within a range of 1–2 mm.

One of the studies for displacement measurements using fiber-optic sensors is presented by Yeon-Gwan Lee et al. [11]. The paper introduces the design of a fiber-optic displacement sensor with a large measurement range. It is composed of a transmissive grating panel, a reflection mirror, and two optical fibers as a transceiver. The measured bidirectional movement demonstrates a peak-to-peak accuracy of 10.5 µm, high linearity of 0.9996 with a resolution of 3.1 µm at the full bandwidth, and a signal-to-noise ratio of 27.7 during a movement of 16 mm.

The performance of the fiber-optic displacement sensor is influenced by its geometrical parameters, such as the fiber aperture, the radius of the fiber core, the lateral separation of the transmitting and receiving fibers, the angle between the two fibers, and the reflector radius. It was observed that for better sensor sensitivity, there should be minimum spacing between transmitting and receiving fibers [12].

The review presented by Chen Zhu et al. [13] illustrated the recent progress of fiber-optic sensors, providing an overview of different physical and mechanical sensors based on this principle. The working principle along with the signal demodulation methods are also shown. Fiber-optic sensors are widely used thanks to their several advantages, such as immunity to electromagnetic interference, corrosion resistance, and small size. On the other hand, the paper presented by Zhilin Xu et al. [14] reported a monolithic dual cavity extrinsic Fabry–Perot interferometer to realize 2D displacement measurement of a target. Two-dimensional random movement detection and the repeatability of the system were investigated experimentally, and demodulation errors better than 96 nm were achieved. This system has many advantages, such as its non-contact characteristics, high accuracy, and compact size, which make it promising to be applied in 2D acceleration measurement. A comparative study of different optimization methods was conducted, and it is explained in the following paragraphs.

For wireless sensor networks (WSNs), where several challenges involve potentially conflicting objectives, satisfying one objective leads to degradation in the other’s performance (if we focus on increasing network lifetime, latency may also increase, which is not desired). So, multi-objective optimization methods are applied to solve this challenge using nature-inspired meta-heuristic algorithms [15]. This method remains more complicated in comparison with the optimization method applied to the sensor in this study. Another study, presented by Wioletta Trzpil et al. [16], proposes a new concept of photoacoustic gas sensing based on capacitive transduction. This method allows full integration while conserving the required characteristics of the sensor. For the sensor performance optimization, a Python programming environment was adapted, and an analytic model was able to find the optimum geometric parameters of a cantilever for photo-acoustic sensing with capacitive transduction. We can see that the geometric parameters can change the sensor performance, which is similar to the optimization method applied in this paper. Another study proposed developing a piezoelectric single-crystal accelerometer with a novel tri-beam structure [17], where a dual objective optimization algorithm is proposed to improve the overall performance; however, this method may be limited to vibration sensors.

A miniature fiber-optic sensor able to provide nanometer resolution over a millimeter range was proposed in the Roberval research laboratory. In two previous studies, the principle of the sensor in one dimension and in two dimensions was respectively validated [9,18]. The objective of this new study is to optimize the performance of the existing fiber-optic displacement sensor regarding its resolution by improving its geometric design parameters. Particularly, the geometric design of the planar reflective grating, in which the unfavorable sensitivity is enhanced, will be focused on.

## 2. Sensor Principle

The sensor consists of two fiber-optic probes associated with a highly reflective surface. Each probe has one center emission fiber and four reception fibers placed around the emission fiber. The sensor performance when it is associated with a planar surface has already been analyzed [9,18,19]. In the classical configuration, the emission fiber placed in the center emits light on a flat reflective surface. The light reflected by the surface is injected into the reception fibers and guided to a PIN photodiode. The voltage output of the sensor is a function of the mirror displacement (see Figure 1). When the flat mirror is translated perpendicular to the probe axis, the sensor response curve is as shown in Figure 2.

As seen in Figure 2, the sensor response curve comprises four zones [14]. The first zone is the dead zone, where the reception fibers cannot collect the reflected light due to the space between the emission and reception fibers. Zones 2 and 4 exhibit strong non-linearity with poor resolution. Zone 3, on the other hand, is the most interesting working zone due to its high sensitivity and linearity. The performance of the sensor is characterized by its sensitivity and resolution in the working zone. The sensitivity (*S*) is calculated as a function of the voltage output variation (Δ*V*) and displacement in the linear zone (Δ*d*) (Equation (1)):(1)$$S=\frac{\Delta V}{\Delta d}$$

The resolution (*R*) is deduced from the sensitivity (*S*) and the RMS noise of the sensor (*N_RMS_*) (Equation (2)):(2)$$R=\frac{N_{RMS}}{S}$$

Nevertheless, zone 3 has a small linear measurement range (<200 µm for OMRON fiber-optics) which is not suitable for long strokes in industrial applications.

To increase the measurement range for the linear zone, the displacement direction of the flat mirror can be different from the normal vector orientation of its surface. This results in the multiplication of the nominal range value by a factor of (sin *ε*)^−1^, where *ε* is the inclination angle related to the grating axis [10], as shown in Figure 3. As a result, the fiber-optic probe displaces laterally to the flat mirror.

In the inclined mirror configuration, the measurement range increases by a factor of (sin *ε*)^−1^, (Equation (3)):(3)$$d_{lateral}=\frac{d_{axial}}{sin\varepsilon }$$
where:-*d_lateral_*: the displacement in the lateral case-*d_axial_*: the corresponding axial displacement

And as *d_lateral_* > *d_axial_*, the sensitivity of the inclined mirror configuration will decrease by a factor of (sin ε), as shown in the following equation:(4)$$S_{lateral}=S_{axial}\times \sin(\varepsilon )$$where:-*S_lateral_*: the sensitivity of the sensor in the inclined mirror configuration (lateral case)-*S_axial_*: the corresponding axial sensitivity

Therefore, the sensor resolution with this inclined mirror configuration increases as a function of the angle *ε* following this equation:(5)$$R_{\varepsilon }=\frac{R}{sin\varepsilon }$$
where:-*R*: the sensor resolution in the classical case-*R_ε_*: the corresponding resolution in the inclined mirror configuration

As seen from the previous equation, the highest resolution (i.e., unfavorable resolution) is obtained with small values of the angle *ε*, whereas for higher values of *ε*, the resolution is improved. To increase the measurement range to several millimeters, the inclined mirror configuration was duplicated, resulting in a grating of flat mirrors. The total displacement of the sensor (*d_total_*) increases as a function of *ε* and the number of steps in the grating (*n*) (Equation (6)):(6)$$d_{total}=n\times \frac{d_{axial}}{sin\varepsilon }$$

In the case of a grating of flat mirrors, two fiber-optic probes are needed to avoid measurement loss due to the transition between two consecutive steps, and that ensures continuous displacement measurement over the long range by alternately switching between the probes (Figure 4). In other words, the non-linear zone in the response curve of the sensor has to be avoided. In order to ensure a useful and correct measurement in this linear zone, two fiber-optic probes are used. When the first one arrives in the non-linear zone, the measurement switches to the next probe. It is noted that the movement of the mirror with respect to the probe can either be to the left or the right.

A geometric model was developed to size the geometric parameters of the grating and to simulate the performances of the long-range displacement sensor. This model takes as an input the geometric dimensions of each fiber and each step of the grating. This model gives the corresponding performance of the sensor as an output. These performances include the sensor resolution and the overlap distance needed to easily switch between the two fiber-optic probes.

Two conditions are taken into account in this model:The distance between the probe and the grating step must be in the linear zone (zone 3 of Figure 2).The overlap distance needed to switch between two successive signals of the fiber-optic probes to avoid the linear measurement discontinuity during the step transition depends on the speed of the measured displacement. It is generally considered to be between 10 and 30 µm.

The algorithm based on the geometric model used to size the planar grating is presented in the following flow chart (Figure 5).

The parameters in the geometric model of the planar grating are shown in the following figure and table (Figure 6 and Table 1).

The geometric model developed in MATLAB calculates the distance (*d*) between the probe and the grating, in addition to the overlap distance necessary to stay in the linear zone and switch between the two probes. In the flow chart shown, we guarantee that the distance (*d*), which is between the probe head and the grating, will be localized in the linear zone of the sensor, and that the overlap has a suitable value between 10 µm and 30 µm. If these conditions are not satisfied, the loop will be ended.

A sensor prototype based on the simulation algorithm was successfully modeled, designed, and tested [10,14].

The following figure shows an illustrative example of the experimental validation for the sensor principle of that prototype. As shown below, two fiber-optic probes are used in order to stay in the linear zone of the sensor.

As seen from Figure 7, there is an overlap of 29 µm to facilitate the switching between the two fiber-optic probes and ensure measurement continuity [14].

In order to improve the sensor’s performance, an optimization method has been proposed. Its aim is to improve the highest sensor resolution (the unsuitable resolution) by reducing its corresponding value. This can be achieved with the help of the geometric parameters, particularly the angle *ε*.

## 3. Optimal Design Approach

The main objective of this design is to determine the optimal dimensions of the sensor’s planar grating, which can improve its resolution; the physical model of the sensor is not yet implemented, and the MATLAB results are based on a program used before where the sensor principle has been validated. Figure 8 shows the classical calibration curve of the fiber-optic displacement sensor for a 300 µm displacement, which is considered in this study [10]. It is observed that increasing the measurement range results in a decrease in sensor sensitivity, as depicted in Figure 9, which shows the instantaneous sensitivity as a function of the sensor displacement. It is evident that the sensitivity reaches its maximum value at the inflection point of the curve, which is found at a displacement of 186 µm and has a maximum sensitivity of 44.28 mV/µm. However, near the inflection point, the sensitivity decreases as the measurement range increases.

The approach followed to reach the optimal performance of the sensor consisted of dividing several zones around the inflection point of the sensor response curve, where each zone has an additional 20 µm length compared to the previous one: (80 µm length for zone 1, 100 µm length for zone 2, 120 µm length for zone 3…etc.). The analysis figured out the zone for which the sensor resolution is optimal. For the overall analysis, six zones were taken around the inflection point, which was sufficient to find the optimal resolution for the sensor.

In each zone, the sensitivity, the measurement range, and the resolution in both the axial and lateral configurations were calculated. Concerning the axial configuration of the sensor, the analysis considered the most unsuitable sensitivity of the measurement range in each zone (the sensitivity at the extremity of the measurement range), from which the maximum axial resolution was deduced (cf. Equation (2)). These values of sensitivity and resolution were the ones considered in this study, with the objective of optimizing the sensor resolution in the worst-case scenario.

Regarding the lateral configuration, the analysis found out the maximum inclination angle (*ε_max_*) in the measurement range of each studied zone to optimize the sensor resolution because the best resolution is attained at a high value of the angle *ε*. For that, and in order to get the highest possible angle, it was necessary to fix a small overlap criterion because, at a small overlap, the angle *ε* is high. For this study, an overlap of 10 µm was taken at each zone as it was the minimum sufficient overlap, providing a high value of the angle *ε*.

So, the approach focused on the minimum sensitivity and, in consequence, the maximum resolution in the axial case (*S_axial min_*, *R_axial max_*) and the maximum angle (*ε_max_*) in the lateral case.

Considering the analysis carried out for zone 1, which has a length of 80 µm around the inflection point (this zone starts at 146 µm and ends at 226 µm), an axial measurement range of 67.5 µm (starting at 158.52 µm and ending at 226 µm) was the one in which the minimum axial sensitivity and maximum resolution were determined (*S_axial min_* = 42.59 mV/µm, *R_axial max_* = 7.04 nm) and the maximum inclination angle (*ε_max_* = 5.54°) was found.

Table 2 presents the different parameters obtained for each zone.

Referring to Equation (2) and taking into consideration the minimal case for the sensitivity (*S_axial min_*); the maximum corresponding resolution will be obtained (*R_axial max_*)

As seen from the previous table, *MR_axial_* and *ε_max_* increase with the zone length. Whereas *S_axial min_* decreases and *R_axial max_* increases.

The parameters that define the sensor performance (*S_axial min_*, *R_axial min_* and *ε_max_*) were used to generate the targeted optimal resolution; this is explained in the next paragraphs.

## 4. Results and Discussion

### 4.1. Analysis of the Optimal Zone

The objective of this study is to define the best resolution for the sensor. It is generated from the parameters previously obtained at each zone.

From (*S_axial min_* and *ε_max_*), the lateral measurement range (*MR_lateral_*) is deduced, and the lateral sensitivity (*S_lateral_*) and the lateral resolution (*R_lateral_*) are obtained, respectively (Equations (7)–(9)):(7)$$MR_{lateral}=\frac{{MR}_{axial\;}}{sin(\varepsilon_{max})}$$
(8)$$S_{lateral\;}=S_{axial\;min}\times \sin\;(\varepsilon_{max})$$
(9)$$R_{lateral}=\frac{R_{axial\;min}}{sin(\varepsilon_{max})}$$

As seen from Table 3, *MR_lateral_* increases with zone length as a function of (sin(*ε*))^−1^. Concerning *S_lateral_* and *R_lateral_*, these two parameters showed their best performance in zone 4 (*S_lateral_* increased to a maximum value at this zone, then it started to decrease; for *R_lateral_*, it decreased to its minimum value in zone 4, then it started to increase).

The previous results proved that there is a global optimum for the sensor in which the lateral sensitivity (*S_lateral_*) and the lateral resolution (*R_lateral_*) were boosted despite enlarging the measurement range, which was not the case in the axial configuration, as axially the sensitivity decreased with the zone range.

*ε_max_* increases as a function of the zone length, and in consequence, the lateral sensitivity and resolution are improved up to a certain limit (zone 4).

As a result, zone 4 is the optimal zone, for which the unsuitable resolution is improved. This zone has a length of 140 µm and a lateral measurement range of 726 µm. The angle *ε_max_* in this zone is 6.25°, which enhances the lateral sensitivity to a maximum value of 4.28 mV/µm and the lateral resolution to a minimum value of 70.32 nm.

The geometric parameters which provided an angle *ε* of 6.25° are:-Step length (*l*) = 1433 µm-Step height (*h*) = 157 µm

These optimal performances were found at the smallest criterion of overlap (10 µm); as at small values of the overlap, higher values of the angle *ε* are obtained, resulting in better resolution.

### 4.2. Study of the Overlap Criterion

The overlap, in general, increases with the step length (*l*), which in consequence decreases the step angle *ε*, and that will deteriorate the sensor lateral resolution (*R_lateral_*), (cf. Equation (5)).

So, increasing the step length (*l*) increases the signal overlap and the limit of resolution for the sensor. Figure 10 presents the results given by the existing geometric model; (Figure 10a) is for the optimal zone where the step length (*l*) is 1433 µm and the corresponding overlap is 10 µm; (Figure 10b) was plotted for a step length equal to 1460 µm, where the overlap increased to 26.5 µm.

The influence of the sensor angle *ε* on the overlap and the resolution was studied in the optimal zone defined in this analysis (Zone 4). For that, the height of the step (*h*) in the geometric model was kept constant at 157 µm, and several values of the step length were applied in order to see how the overlap and the resolution change with the angle *ε* (Figure 11).

Figure 11 shows that the resolution is proportional to the step angle of the grating (*ε*), as proved before, whereas the overlap is inversely proportional to the angle *ε*, which means that at high values of overlap, the sensor resolution is not optimized. For that reason, in this analysis, a minimum criterion was considered for the overlap (10 µm) to optimize the sensor performances regarding the resolution.

On the other hand, the overlap criterion is related to the velocity of the measurement system and its sampling frequency. It is necessary to have enough measured points in the overlap zone in order to facilitate the switching between the two fiber-optic probes. Table 4 presents the number of points in the overlap zone of 10 µm at different velocities.

As seen from Table 3, with an overlap of 10 µm and sampling frequencies (100–200 Hz), there will be a smaller number of points in the overlap zone. In that case, the overlap criterion should be increased (Table 5).

With an overlap of 30 µm, a velocity of 0.2 mm/s, and sampling frequencies (100–200 Hz), the number of points in the overlap zone is increased to 15 and 30 points, respectively, which is more optimal for a better functionality of the sensor (with a high number of points, the precision is improved). For the velocity at 2 mm/s, the overlap criterion should be further increased at sampling frequencies of 100–200 Hz.

## 5. Conclusions

The geometric design of a fiber-optic displacement sensor is enhanced regarding its sensitivity, resolution, and measurement range. In this paper, a global optimum is generated between the sensor sensitivity and resolution, which, in consequence, improves its overall performance. This global optimum has laterally enhanced the sensitivity and the resolution, even if axially the performance was in its unfavorable case; this has been carried out with the help of the angle *ε*, which was chosen to be at its maximum value.

The following approach proved its validity as the sensitivity of the sensor increased to 4.28 mV/µm, despite enlarging the measurement range. However, higher values of sensitivity could have been reached if, axially, the performances were better. On the other hand, a suitable overlap criterion should be considered as a function of the measurement system’s velocity and the sampling frequency.

The geometric parameters for the sensor at its optimal zone will be considered for future fabrication of the grating to validate experimentally this global optimum. In addition, here is a comparative table of several sensors. With the sensor mentioned in this study, in terms of resolution and measurement range, it can be seen that even in the worst-case scenario, the limit of resolution is nanometric (Table 6).

## Figures and Tables

**Figure 1 sensors-24-00448-f001:**
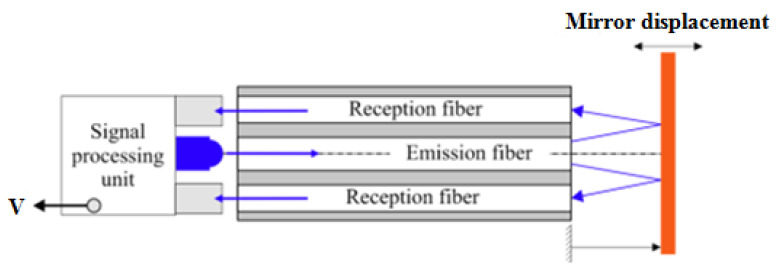
Fiber-optic sensor.

**Figure 2 sensors-24-00448-f002:**
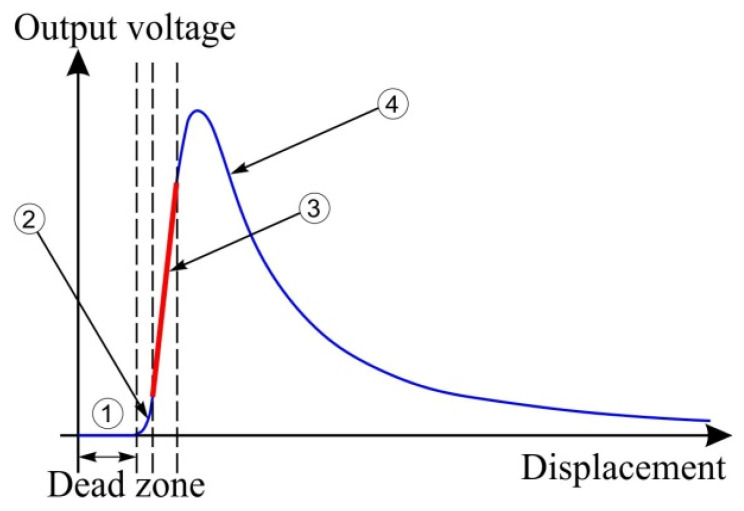
Response curve of the fiber-optic displacement sensor.

**Figure 3 sensors-24-00448-f003:**
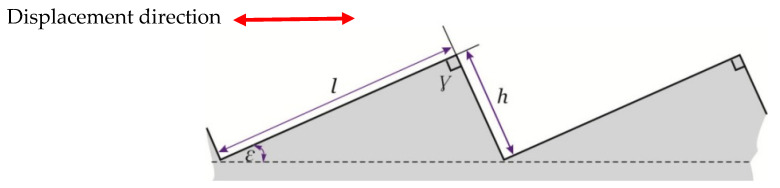
Inclined mirror configuration.

**Figure 4 sensors-24-00448-f004:**
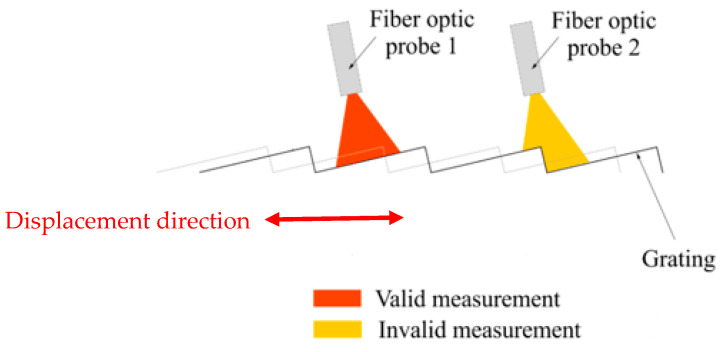
Long-range sensor principle.

**Figure 5 sensors-24-00448-f005:**
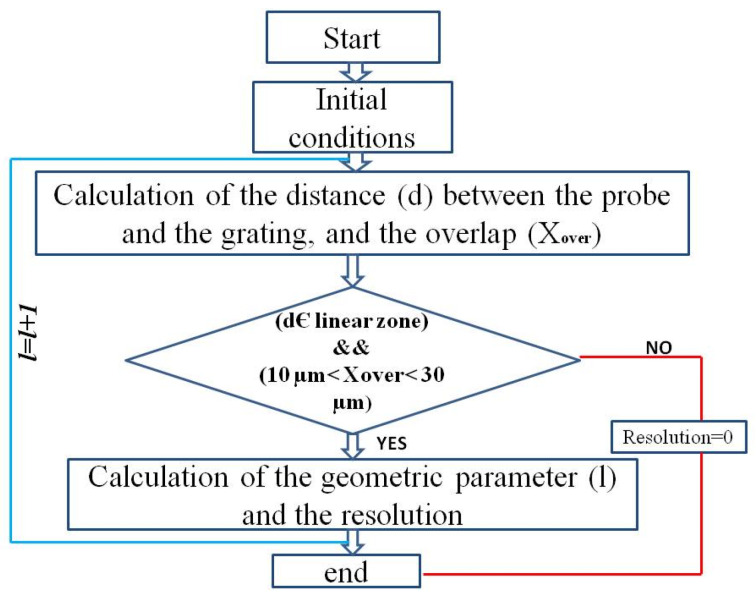
Flow chart of the geometrical model.

**Figure 6 sensors-24-00448-f006:**
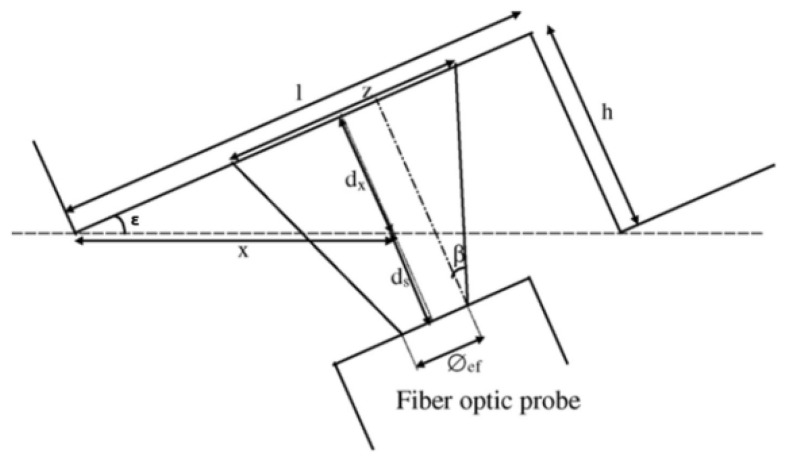
Grating and emission fiber parameters.

**Figure 7 sensors-24-00448-f007:**
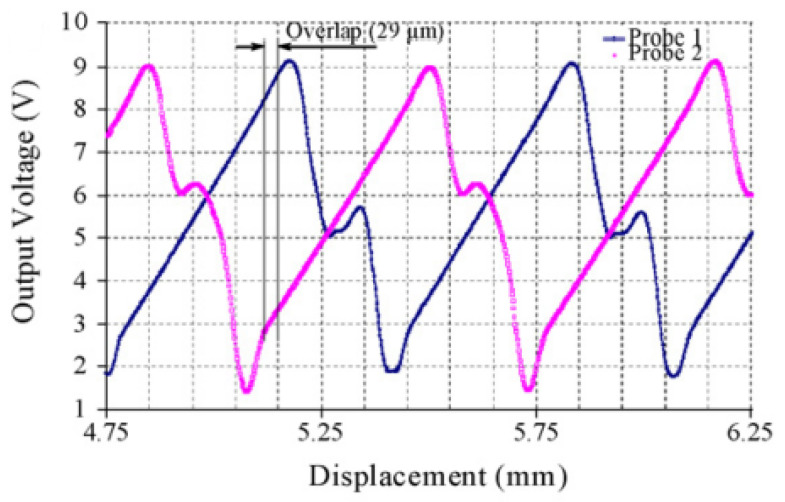
Long-range measurement.

**Figure 8 sensors-24-00448-f008:**
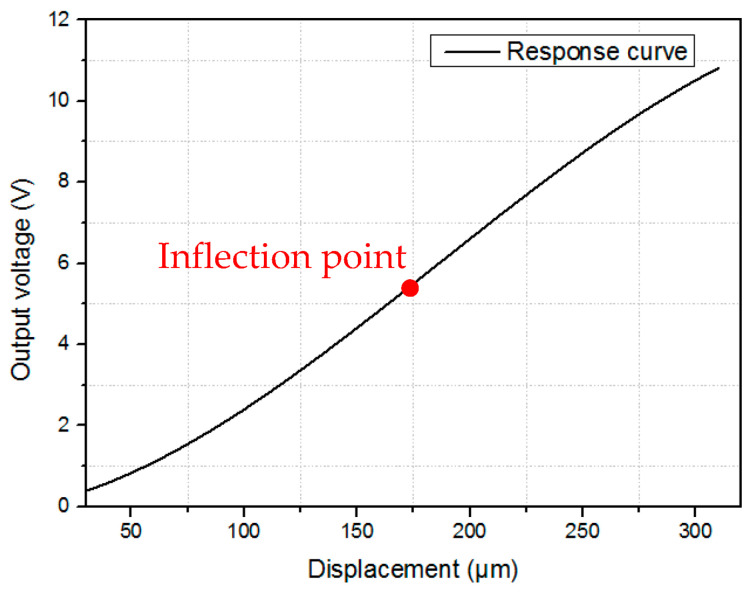
Calibration curve of the sensor.

**Figure 9 sensors-24-00448-f009:**
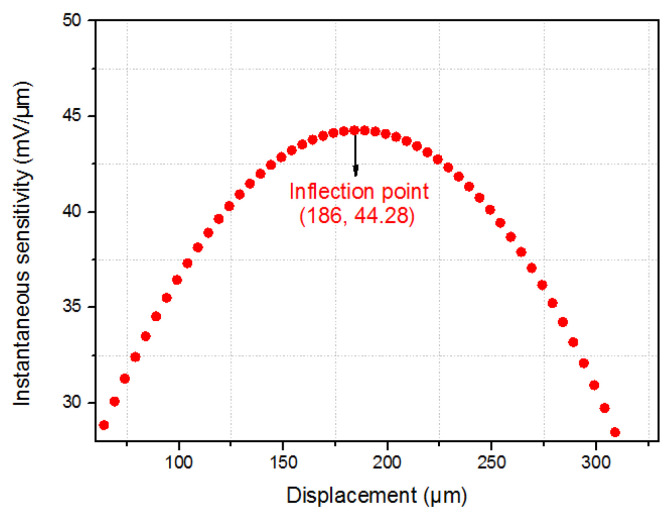
Instantaneous sensitivity (mV/µm) as a function of displacement (µm).

**Figure 10 sensors-24-00448-f010:**
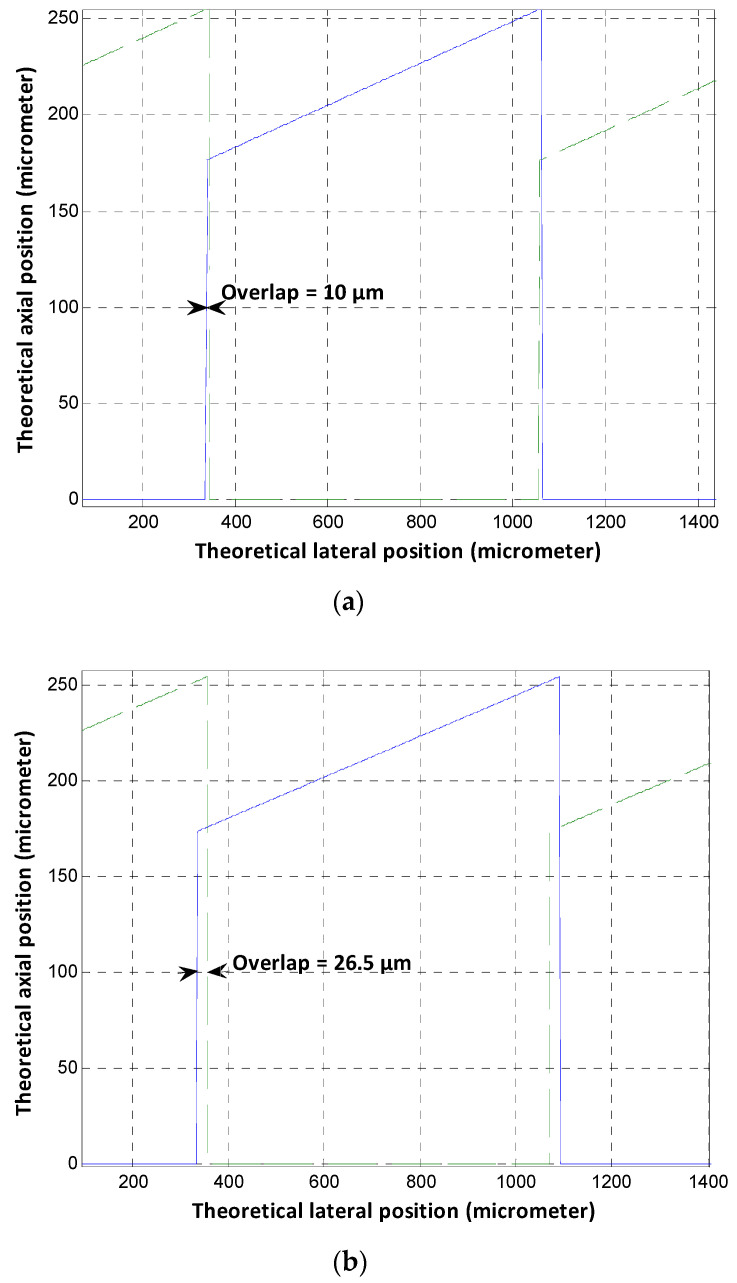
Geometric model results. (**a**) *l* = 1433 µm, *ε* = 6.25°, (**b**) *l* = 1460 µm, *ε* = 6.14°.

**Figure 11 sensors-24-00448-f011:**
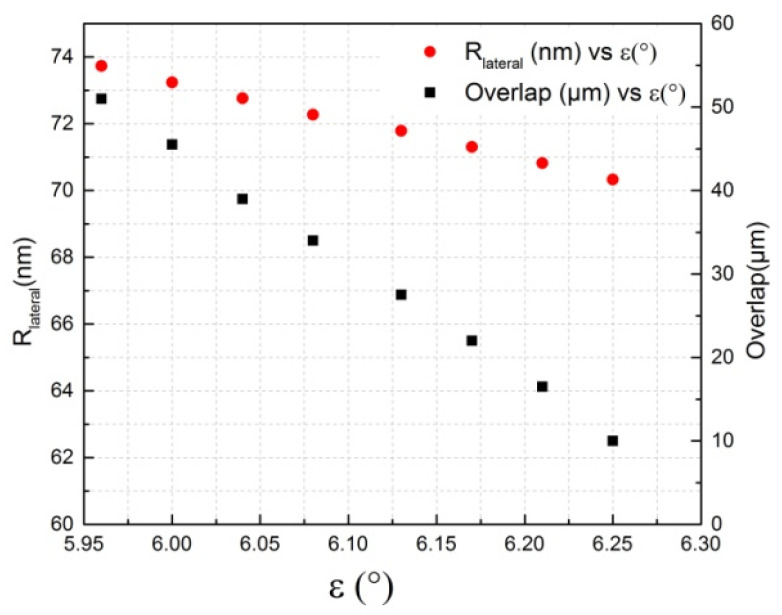
Overlap (µm) vs. angle (°).

**Table 1 sensors-24-00448-t001:** Geometric design parameters.

Symbol	Quantity
*Φ_ef_* (µm)	Emission fiber diameter
*Φ* (µm)	Probe diameter
*β*	Emission fiber numerical aperture
*d* (µm)	Distance between probe head and grating
*d_x_* (µm)	Distance: *x*·sin(*ε*)
*d_s_* (µm)	Security distance
*ε* (°)	Grating angle
*x* (µm)	Lateral position
*l* (µm)	Step length
*h* (µm)	Step height
*z* (µm)	Illuminated zone diameter

**Table 2 sensors-24-00448-t002:** Optimal design parameters.

Zone	Zone Length (µm)	*MR_axial_* (µm)	*S_axial min_* (mV/µm)	*R_axial max_* (nm)	*ε_max_* (°)
1	80	67.5	42.59	7.04	5.54
2	100	71	41.65	7.2	5.76
3	120	75.09	40.51	7.4	6.01
4	140	79.02	39.15	7.66	6.25
5	160	83.11	37.58	7.98	6.50
6	180	87.14	35.81	8.38	6.73

where: *MR_axial_*: the axial measurement range; *S_axial min_*: the minimum axial sensitivity; *R_axial max_*: the maximum axial resolution; *ε_max_*_:_ the maximum angle.

**Table 3 sensors-24-00448-t003:** Optimal design results.

Zone	Zone Length (µm)	*MR_lateral_* (µm)	*S_lateral_* (mV/µm)	*R_lateral_* (nm)
1	80	700	4.11	72.98
2	100	709	4.19	71.77
3	120	718	4.25	70.73
**4**	**140**	**726**	**4.28**	**70.32**
5	160	735	4.27	70.51
6	180	744	4.22	71.44

**Table 4 sensors-24-00448-t004:** Overlap criterion of 10 µm.

Velocity (mm/s)	Overlap (µm)	Sampling Frequency (Hz)	Number of Points
0.2	10	100	5
0.2	10	200	10
2	10	100	0.5
2	10	200	1

**Table 5 sensors-24-00448-t005:** Overlap criterion of 30 µm.

Velocity (mm/s)	Overlap (µm)	Sampling Frequency (Hz)	Number of Points
0.2	30	100	15
0.2	30	200	30
2	30	100	1.5
2	30	200	3

**Table 6 sensors-24-00448-t006:** Comparative between high-performance displacement sensors.

Sensor	Range	Resolution
Conventional Fabry–Perot interferometer [3]	100 mm	Quarter wavelength
LDGI [4]	10 mm	30 nm
Optical sensor of laser diode module [7]	Centimetric	Nanometric
FBG [10]	1–2 mm	0.48 µm
Fiber-optic sensor [11]	16 mm	3.1 µm
Dual cavity Fabry–Perot interferometer [14]	7 µm	0.1 nm
Fiber-optic sensor of this study	726 µm	70.32 nm (in worst-case scenario)

## Data Availability

Data are contained within the article.

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
