# Peer review of "Optimizing Algorithm for Existing Fiber-Optic Displacement Sensor Performance"

_sensors, 2024, doi:10.3390/s24020448_

Round 1
Reviewer 1 Report
Comments and Suggestions for Authors
In this paper, the authors investigated a fiber optic displacement sensor. The sensor itself was proposed decades ago as one classic fiber sensor based on light intensity, the authors did some detailed analyses and optimization into the design of ‘the mirror’, which is a grating structure in this work.
I would strong advise them to compare their work with other state-of-the-art methods and present more experimental results from different angles, to build a solid base for their contributions before this paper could be considered to be published. Many of the figures are also not clear enough, with text in the figure hard to read.
Author Response
Dear reviewer,
In response to your comment, different optimization methods have been investigated in the state of art. For wireless sensor networks (WSN) , where several challenges involve potentially conflicting objectives ; satisfying one objective leads to degradation in other's performance (if we focus on increasing network lifetime, latency may also increase, which is not desired). So, the multi-objective optimization methods are applied to solve this challenge using nature inspired meta-heuristic algorithms. This method remains more complicated in comparison with the optimization method applied for the sensor in this study. Another study, presented by Wioletta Trzpil et. al, where a new concept of photoacoustic gas sensing based on capacitive transduction is proposed, this method allows full integration while conserving the required characteristics of the sensor, for the sensor performance optimization , a python programming environment was adapted , where an analytic model has been able to find the optimum geometric parameters of a cantilever for photo-acousting sensing with capacitive transduction , we can see that the geometric parameters can change the sensor performance , which is similar to the optimization method applied in this paper. Another study, proposed developing a piezoelectric single-crystal accelerometer with a novel tri-beam structure, where a dual objective optimization algorithm is proposed to improve the overall performance, however , this method maybe limited to vibration sensors.
Experimental results will be the objective of future works in the laboratory; as, theoretically, this method proved its own feasibility.
The sizes of many figures have been adapted , so it is cleared to visualise.
Reviewer 2 Report
Comments and Suggestions for Authors
The manuscript titled “Optimal Design and Performances Enhancement of a Fiber-Op- 2 tic Displacement Sensor” reports the design of a fiber optic displacement sensor based on a triangular reflective grating and two optical fiber probes.
1) The literature review is not complete. There are several important references are missing. Examples include the following refs.
Zhu, C., Zheng, H., Ma, L., Yao, Z., Liu, B., Huang, J. and Rao, Y., 2023. Advances in Fiber-Optic Extrinsic Fabry-Perot Interferometric Physical and Mechanical Sensors: A Review. IEEE Sensors Journal.
Xu, Z., Wang, Z., Chen, L., Fan, J., Tu, L. and Zhou, Z., 2021. Two-dimensional displacement sensor based on a dual-cavity Fabry-Perot interferometer. Journal of Lightwave Technology, 40(4), pp.1195-1201.
2) Fig. 2 and Fig. 3 are basically the same. It is suggested to combine these two in one figure.
3) It is suggested to experimentally investigate the dynamic range and resolution of an optimized sensor design, to really demonstrate the proposed strategy.
Author Response
Dear reviewer,
In response to your comment, different optimization methods have been investigated in the state of art and added into the introduction, in addition to the two suggested references.
The review presented by Chen Zhu et. al. , illustrated the recent progress of fiber-optic sensors, providing an overview of different physical and mechanical sensors based on this principle. The working principle along with the signal demodulation methods are also shown. The fiber-optic sensors ae widely used , thanks to their several advantages, such as : immunity to electromagnetic interference, corrosion resistance and small size. On the other hand, the paper presented by Zhilin Xu et. al, reported a monolithic dual cavity extrinsic fabry-perot interferometer to realize 2D displacement measurement of a target. 2D random mouvement detection and the repeatability of the system were investigated experimentally , where demodulation errors better than 96 nm were achieved, this system has many advantages, such as: its non-contact characteristics, high accuracy and compact size, which make it promising to be applied in the 2D acceleration measurement. A comparative study of different optimization methods have been done, and it is explained in the following paragraphs:
For wireless sensor networks (WSN) , where several challenges involve potentially conflicting objectives ; satisfying one objective leads to degradation in other's performance (if we focus on increasing network lifetime, latency may also increase, which is not desired). So, the multi-objective optimization methods are applied to solve this challenge using nature inspired meta-heuristic algorithms. This method remains more complicated in comparison with the optimization method applied for the sensor in this study. Another study, presented by Wioletta Trzpil et. al, where a new concept of photoacoustic gas sensing based on capacitive transduction is proposed, this method allows full integration while conserving the required characteristics of the sensor, for the sensor performance optimization , a python programming environment was adapted , where an analytic model has been able to find the optimum geometric parameters of a cantilever for photo-acousting sensing with capacitive transduction , we can see that the geometric parameters can change the sensor performance , which is similar to the optimization method applied in this paper. Another study, proposed developing a piezoelectric single-crystal accelerometer with a novel tri-beam structure, where a dual objective optimization algorithm is proposed to improve the overall performance, however , this method maybe limited to vibration sensors.
- Figure 3 is removed and the explanation is based on figure 2 only.
- Experimental results will be the subject of future works in the laboratory, as this method proved its feasibility theoretically.

Reviewer 3 Report
Comments and Suggestions for Authors
Dear Colleagues!
Your rather interesting article is devoted to solving one of the most difficult problems of scientific practice - the optimization problem. The goal of your work is to optimize the performance of the existing fiber-optic displacement sensor regarding its resolution by improving its geometric design parameters.
Based on this, I would like to understand your vision and get answers to some aspects of the article.
1. You have provided a fairly rich overview in the introduction. However, it does not allow reader to understand the premises that allowed you to formulate the specified purpose of the work. The huge number of given characteristics does not allow, however, to understand their connection with the geometric design parameters of existing sensors. Lines 27-30 and 62-66 repeat each other or the difference is not emphasized in them. As a recommendation, you can propose a summary of the introduction, presented in the form of a table - a matrix for searching new solutions based on an analysis of the advantages and disadvantages of existing methods and the connection of their characteristics with the geometry of the sensors.
2. From the second section it is not clear what kind of modernization of the geometry of the sensor itself we are talking about. Judging by Fig. 4, you only changed the direction of movement of the sensor in the old geometry relative to the object being studied. At the same time, Fig. 4b is somewhat puzzling, since in the specified configuration and direction of movement the sensor will collide with the grating.
3. In addition, the geometry of Fig. 7 requires clarification on reflected light. Otherwise, it is not clear how the determination of the correctness or incorrectness of measurements with overlapping grating steps is realized.
4. Further in the second section the phrase flashes that you are using two sensors, not one. Page 189-190 "As shown below, two fiber-optic probes are used, in order to stay in the linear zone of the sensor." What does it mean?
5. And finally, I would like to see an analysis of the influence of the environment on the accuracy of measurements; a comparison table indicating where the improved performance of your sensor compares to others (preferably in the same units, for example nm, and not thousands of pm, as you indicated for your sensor); clarification of a precise understanding of what it means to optimize the design of your sensor or, indeed, to optimize the measurement circuit.
Based on the answers to these questions, the reaction to these recommendations, and the clarification of a number of aspects of the article, a decision will be made on the possibility of its publication.
Author Response
- Lines 27-30 have been deleted, and other optimization methods in the state of art have been investigated and added to the introduction.
- The figure is just a schematic diagram , showing the displacement direction , moreover the motion is relative , we fix the probe and the grating moves, where the probe does not collide the grating.
- Firstly, the minimum diameter z is limited by the emission fiber
core diameter φef. Secondly,z depends on the numerical aperture
of the emission fiber, nair·sin β (nair ≈ 1), and on the distance d
between the probe head and the grating tooth. This distance can
be divided into two parts: dx = x sin α depending on the probe
position x and ds which is a security distance to avoid collisions
between the probe and the grating during the movement.
So, the limit condition for z is: z < l (length of a grating tooth)
because the probe cannot illuminate more than one tooth otherwise the displacement measure is lost. Since z is limited, ds cannot be too long. That's what I can give as explanation for figure 7, can the reviewer be more clear? - We have to avoid the non linear zone in the response curve of the sensor , to ensure that we always get measurement in this linear zone, we use two probes, when the first one arrives the non-linear zone, the measurement switches to the next probe.
- Can the reviewer specifies which type of technology or sensors he would like to have comparative study.
Reviewer 4 Report
Comments and Suggestions for Authors
The authors propose the design optimization of a miniature fiber-optic linear displacement sensor with the purpose of enhancing its performance, specifically its resolution. To do so, the authors use a miniature fiber-optic sensor previously reported by the group. It's hard to understand how the sensor was optimized, what is the design difference concerning previous works. Also, the authors say that the resolution has been improved, but they don't compare it with previous results. I don’t find coherence in the presented work or innovation (recall [10] and [14]) that would be worth to publish in MDPI Sensors Journal.
Note that:
Figures 2, 3 and 7 come from previously reported work [10].
Figures 5 and 8 come from previously reported work [14].
Author Response
The objective of this study is to define the best resolution for the sensor. It is generated from the parameters previously obtained at each zone taken arounf the inflection point.
From (Saxial min and εmax), the lateral measurement range (MRlateral) is deduced, the lateral sensitivity (Slateral) and the lateral resolution (Rlateral) are obtained, respectively , which means we focused on the minima axial resolution along with the highest angle to have optimum between the two parameters.
Round 2
Reviewer 1 Report
Comments and Suggestions for Authors
This time the authors have made a few changes to the paper and the figures are more clear.
It would be a good advice to revise the title of the paper to include their algorithm to promote their contributions. And as in the current form, it seems that the research spotlight is on the intensity displacement fiber optical sensor, which is not novel as it has been worked on dozens of years ago by other researchers.
And by comparing different methods and sensors, I meant theoretical and experimental comparisons between this sensor and other sensors, so that the readers could know that your sensor along with your algorithm work better.
Author Response
Response to Reviewer 1
- This time the authors have made a few changes to the paper and the figures are more clear.
Thank you very much.
- It would be a good advice to revise the title of the paper to include their algorithm to promote their contributions. And as in the current form, it seems that the research spotlight is on the intensity displacement fiber optical sensor, which is not novel as it has been worked on dozens of years ago by other researchers.
Actual title: Optimal Design and Performances Enhancement of a Fiber-Optic Displacement Sensor.
Proposed new title: Optimizing Algorithm for Existing Fiber-Optic Displacement Sensor Performance. (It is just a suggestion where we highlight on algorithmic approach, and the fact that the sensor is an existing one), this title has been added to the actual revised manuscript.
- And by comparing different methods and sensors, I meant theoretical and experimental comparisons between this sensor and other sensors, so that the readers could know that your sensor along with your algorithm work better.
A comparison along with other sensors has been identified in table 1:
|
Sensor |
Range |
Resolution |
|
Conventional Fabry-Perot Interferometer [3] |
100 mm |
Quarter wavelength |
|
LDGI [4] |
10 mm |
30 nm |
|
Optical Sensor of Laser Diode Module [7] |
centimetric |
nanometric |
|
FBG [10] |
1-2 mm |
0.48 µm |
|
Fiber-Optic Sensor [11] |
16 mm |
3.1 µm |
|
Dual Cavity Fabry-Perot Interferometer [14] |
7 µm |
0.1 nm |
|
Fiber-Optic sensor of this study |
726 µm |
70.32 nm (in worst case scenario) |
However, experimental work on this sensor combined with this new algorithm has not been done yet, it will be a part of future works. It can be seen from the above table that this sensor along with this algorithm, can give a resolution of 70.32 nm in its worst-case scenario, as the analysis has considered it, so, this resolution can be better also to another cases.

Reviewer 3 Report
Comments and Suggestions for Authors
Dear Colleagues!
I apologize for some delay due to Covid.
After reading your answers, I have to say the following.
According to p. 1. Questions removed.
Regarding p. 2. The explanation you gave does not remove the essence of the issue. Fig. 4b and fig. 5 differ significantly in motion geometry, although they are the same thing. Therefore, I ask you to once again pay attention to the direction of movement in Fig. 4b.
According to p. 3. This geometry is clear, as evidenced by your answer. But it does not correspond to the one shown in Fig. 1. In Fig. 7 does not show the course of reflected rays.
Both questions under pp. 2 and 3 are aimed at some clearer display of similar situations, so that the reader can understand what is being said, even without reading the text, but by looking at the pictures. That is, each subsequent figure, for example, 5, follows from Fig. 4b, a fig. 7 corresponds to Fig. 1, but in a new sensor configuration. Thus, the figures have to be likely the same in important places.
Regarding p. 4, I would like to see this explanation in the text of the article.
According to p. 5. The updated review contains works that provide data on the metrologic characteristics of sensors similar in purpose of measuring - displacement. I would like to see a comparison table of this data with the data obtained in your work, for example 3-4 lines plus your article.
In conclusion, I ask you to pay attention to the low quality of most of the drawings.
Author Response
Dear reviewer,
Please find the point-by-point reply in the attachment.

Reviewer 4 Report
Comments and Suggestions for Authors
Dear authors,
It was expected to see a proper review of the paper. Instead, only the introduction was improved. Figure 3 was removed and captions of following figures were not updated.
Also, I would like to see a clarification of the concerns that I previously raised. Let's recall them again:
Also, the authors say that the resolution has been improved, but they don't compare it with previous results [10] and [14].
Please explain the following:
Figures 2, 3 and 7 come from previously reported work [10].
Figures 5 and 8 come from previously reported work [14].
Considering the paper in its current state, I do not recommend its publication.
Author Response
Response to Reviewer 4
- It was expected to see a proper review of the paper. Instead, only the introduction was improved. Figure 3 was removed and captions of following figures were not updated.
Some changes have been done to the manuscript, in the introduction part, a comparative table study of different sensors has been added, the captions of all the tables and their recall have been updated. Figure 4 has been replaced by another figure to show the inclined mirror configuration and the displacement direction, an explanation of the use of two probes have been added. The caption of the figures has been updated and shown in red.
- Also, the authors say that the resolution has been improved, but they don't compare it with previous results [10] and [14].
The studies mentioned in [10], which now reference [9] in the revised manuscript and [14], which is now [18] have been done in a very early stage (in 2006 & 2010) , where the electronic noise of the sensor was very low, and therefore, the resolution was very low ; for [9] it is about 14.5 nm and in [18] it is about 38.7 nm, this analysis has been done on the same fiber-optic probe but with a generation of another response curve, so, the electronic noise is higher than the one in the studies of [9] and [18], that’s why we arrive at a resolution of about 70.32 nm (if this analysis has been done on the response curve of the studies of [9] and [18] , it would have been better also, but not for the actual sensor.
- Please explain the following: Figures 2, 3 and 7 come from previously reported work [10]. Figures 5 and 8 come from previously reported work [14].
This optimization method has been done on the fiber-optic probe associated to an inclined mirror grating configuration, it is very important for the reader to understand the complete structured history of the sensor, for him/her to understand the optimization method which is based upon the inclined angle (ε). And the recall of these figures allow the reader to understand the measurement principle in detail, however I obtained the copyright permission to use these figures (the use of these figures has the intention for a better understanding of the measurement principle of the sensor).
Round 3
Reviewer 3 Report
Comments and Suggestions for Authors
Dear Colleagues!
Our progress towards success seems positive to me.
The issue regarding point 2 has been resolved.
Question about point 3. Has the design of your sensor changed or not?
If so, please provide a drawing of the new design.
If not, note that in fig. 1 you show two trajectories for the diverging beam of the emitting fiber, and the reflected rays entering the receiving fibers. In contrast to this in Fig. 4 you show only diverging rays, no reflected ones. In Fig. 6 is the same. In this regard, the question arose whether your sensor design had changed. And what part of light power are gathered by receiving fibers? May be parasitic light from mirror form are gathered also?
Question on point 4. For simplicity, please clarify the illustration of the direction of displacement. In Fig. 3 it is pointed to the right. In Fig. 4 it is pointed to the left. Does the direction of displacement matter? Indicate your opinion on this issue in the article. Or write that the displacement occurs in one direction and make the arrows unidirectional. Or write that movement can be either left or right, and make the arrows bi-directional.
The issue under item 5 has been resolved.
Question on item 6. Thank you for listening to my opinion and making a comparison table. But according to elementary logic, it should be in the results section or in the conclusion to emphasize the importance and effectiveness of your work. But, It was placed by you above, in the introduction, which gives the impression of prematureness until the reader has become familiar with the results of the work.
The issue regarding item 7 has been resolved.
Author Response
Response to Reviewer 3
- Our progress towards success seems positive to me.
Thank you very much.
.
- The issue regarding point 2 has been resolved.
Thank you very much.
- Question about point 3. Has the design of your sensor changed or not? If so, please provide a drawing of the new design. If not, note that in fig. 1 you show two trajectories for the diverging beam of the emitting fiber, and the reflected rays entering the receiving fibers. In contrast to this in Fig. 4 you show only diverging rays, no reflected ones. In Fig. 6 is the same. In this regard, the question arose whether your sensor design had changed. And what part of light power are gathered by receiving fibers? May be parasitic light from mirror form are gathered also?
The design presented in figure 1, is the initial design of the sensor where the mirror moves perpendicularly to the probe. This has been shown to illustrate the measurement principle of the sensor. Figure 4 has been changed. This design has been gradually changed, where the mirror moves laterally to the probe (in the inclined mirror configuration). The inclined mirror configuration is the one used in the study along with the optimization algorithm. In figure 6, the illuminated zone diameter z is defined as follows: z = φef + 2d tan β. Firstly, the minimum diameter z is limited by the emission fiber core diameter φef. Secondly, z depends on the numerical aperture of the emission fiber, nair·sin β (nair ≈ 1), and on the distance d between the probe head and the grating step. This distance can be divided into two parts: dx = x sin α depending on the probe position x and ds which is a security distance to avoid collisions between the probe and the grating during the movement. So, the limit condition for z is: z < l (length of a grating tooth) because the probe cannot illuminate more than one step, otherwise the displacement measure is lost. Since z is limited, ds cannot be too long. However, ds cannot be chosen as small as desired because the probe diameter φ imposes an inferior limit on distance ds :ds > φ 2 tan ε.
- Question on point 4. For simplicity, please clarify the illustration of the direction of displacement. In Fig. 3 it is pointed to the right. In Fig. 4 it is pointed to the left. Does the direction of displacement matter? Indicate your opinion on this issue in the article. Or write that the displacement occurs in one direction and make the arrows unidirectional. Or write that movement can be either left or right and make the arrows bi-directional.
The displacement direction can be bi-directional, either to the left or to the right. The arrows are changed in the figures 3 & 4 to be bi-directional.
- The issue under item 5 has been resolved.
Thak you very much.
- Question on item 6. Thank you for listening to my opinion and making a comparison table. But according to elementary logic, it should be in the results section or in the conclusion to emphasize the importance and effectiveness of your work. But, It was placed by you above, in the introduction, which gives the impression of prematureness until the reader has become familiar with the results of the work.
The table has been removed from the introduction part, and replaced in the conclusion part.

Reviewer 4 Report
Comments and Suggestions for Authors
Dear Authors
I understand and accept the answers to my concerns. The paper may be accepted for publishing.
Author Response
Dear reviewer,
Thank you very much.